# Understanding the Intricacies of Iron Overload Associated with β-Thalassemia: A Comprehensive Review

**Subhangi Basu** [1], **Motiur Rahaman** [1], **Tuphan Kanti Dolai** [2], **Praphulla Chandra Shukla** [1] and **Nishant Chakravorty** [1,*]

1    School of Medical Science and Technology, Indian Institute of Technology Kharagpur, Paschim Medinipur, Kharagpur 721302, West Bengal, India; subhangi9@kgpian.iitkgp.ac.in (S.B.); motiurrahaman24@gmail.com (M.R.); pcshukla@smst.iitkgp.ac.in (P.C.S.)

2    Department of Hematology, Nil Ratan Sircar Medical College and Hospital, Kolkata 700014, West Bengal, India; tkdolai@hotmail.com

*    Correspondence: nishant@smst.iitkgp.ac.in

**Abstract:** β-thalassemia, a congenital genetic hematological disorder characterized by the decrease or absence of β-globin chains, leads to a decrease in levels of Hemoglobin A. The affected individuals can be categorized into two cohorts based on transfusion dependency: transfusion-dependent thalassemia (TDT) and non-transfusion-dependent thalassemia (NTDT). Remarkably, despite the primary pathology lying in β-globin chain depletion, β-thalassemia also exhibits an intriguing association with iron overload. Iron metabolism, a tightly regulated physiological process, reveals a complex interplay in these patients. Over time, both cohorts of β-thalassemic individuals develop iron overload, albeit through distinct mechanisms. Addressing the diverse complications that arise due to iron overload in β-thalassemic patients, the utilization of iron chelators has gained a lot of significance. With varying efficacies, routes of administration, and modes of action, different iron chelators offer unique benefits to patients. In the Indian context, three commercialized iron chelators have emerged, showcasing a high adherence rate to iron chelator-based treatment regimens among β-thalassemic individuals. In this review, we explore the intriguing connection between β-thalassemia and iron overload, shedding light on the intricate mechanisms at play. We delve into the intricacies of iron metabolism, unveiling the distinct pathways leading to iron accumulation in these patients. Additionally, the therapeutic efficacy of different iron chelators in managing iron overload complications is mentioned briefly, along with the guidelines for their usage in India. Through this comprehensive analysis, we aim to deepen our understanding of β-thalassemia and iron overload, paving the way for optimized treatment strategies. Ultimately, our findings provide valuable insights into improving the care and outcomes of individuals affected by β-thalassemia.

**Keywords:** β-thalassemia; iron overload; iron chelators; TDT; NTDT

## 1. Introduction

Thalassemia syndromes are reported as a cluster of multi-genetic inherited hematological diseases that develop due to impaired formation of one or more chains of hemoglobin [1]. Globally, around 56,000 infants are born with severe thalassemia (alpha or beta) annually, with more than half of them reported to require regular blood transfusions [2]. β-thalassemia is represented by decreased (β⁺) or absent (β⁰) synthesis of β-globin chains of the most prevalent form of adult hemoglobin, Hemoglobin A ($\alpha_2\beta_2$), due to one or more mutations in the intronic, exonic, and/or promoter region of β-globin (HBB) genes, which are present on chromosome 11 [3,4]. According to the previous data available, β-thalassemia has been described to primarily occur as an autosomal recessive disorder. β-thalassemic individuals can be divided into three cohorts: β-thalassemia major (TM), β-thalassemia intermedia (TI), and β-thalassemia minor (carrier) [5]. It was estimated that around 10,000–12,000 TM infants are born yearly in India, and around 42 million

β-thalassemia carriers are present in India. The annual prevalence rate of β-thalassemia was determined to be around 3–4% [6,7]. Due to the quantitative reduction in β-globin, particularly in individuals with TM and TI, excess accumulation of α-globin chains in erythroid precursors has been reported. This causes globin chain imbalance, resulting in a state called 'ineffective erythropoiesis'. Under such circumstances, in an effort to enhance erythrocyte production (red blood cells), the developing nucleated erythroid cells undergo premature apoptosis as a means to restore equilibrium. This ultimately leads to chronic hemolytic anemia, which requires regular blood transfusions and other therapeutic approaches, like iron chelation therapies, fetal hemoglobin upregulation, etc., to alleviate disease symptoms, as well as other clinical approaches, like hematopoietic stem cell transplantation, etc., to resolve the disease pathology [8–10]. For the classification of thalassemic disorders, dependence on blood transfusion has also been considered as a parameter; hence, it is considered that there are two types of thalassemia: TDT (transfusion-dependent thalassemia) and NTDT (non-transfusion-dependent thalassemia) [11]. TDT patients must obtain lifelong, recurrent blood transfusions, whereas NTDT patients require occasional or infrequent blood transfusions.

Iron is a biometal that is reported as a crucial micronutrient for the survival, growth, and sustenance of all organisms, involved in various significant biological processes like cellular proliferation, certain redox reactions, cell cycle progression, DNA synthesis, ferroptosis, etc. [12,13] It is a cofactor of multiple enzymes because of its capacity to form complexes with organic ligands [14]. An average human is known to maintain a reserve of 3–5 g of iron under physiological conditions (around 55 mg/kg in males and around 44 mg/kg in females), differentially dispersed across various cell types [15]. Around 80% of the iron pool in the human body is related to the hemoglobin present in red blood cells, whereas the rest is contained in macrophages and liver hepatocytes [16]. The fine balance of the iron level in the human body, as maintained by iron metabolism, is critical for homeostasis. Any disequilibrium on either side, leading to deficiency or overload, has been linked with cellular damage and damage to various organs in the body. Remarkably, iron overload is frequently reported as a major consequence of β-thalassemia (both TDT and NTDT). Transfusion-dependent β-thalassemic (TDT) individuals who receive regular blood transfusions are predisposed to secondary iron overload in diverse organs, like the liver, heart, etc., and have a greater propensity towards the development of iron toxicity [17]. In NTDT, ineffective erythropoiesis primarily leads to iron overload in patients. Ineffective erythropoiesis in β-thalassemia induces an elevated production of erythroid progenitor cells, and this consequently requires increased intestinal iron absorption, which ultimately gets deposited in different organs of the body, instead of aiding in the formation of more erythrocytes [10,18,19]. Furthermore, ineffective erythropoiesis results in increased serum erythropoietin levels, and this is coupled with a decline in serum hepcidin levels (detailed mechanism provided in Section 3.1. TDT vs. NTDT), which ultimately results in enhanced iron uptake and, eventually, iron overload in different organs, as well [20,21]. The absence of an effective mechanism for the elimination of excess iron from the human body, especially in such conditions, leads to a plethora of comorbidities, associated with TDT and NTDT as repercussions of iron overload [22].

This review explores the various mechanisms of iron overloading in β-thalassemic individuals and discusses the molecular pathways involved in such iron overload conditions. Subsequently, we take a critical look at commercially available iron chelators and the contemporary scenario of their usage in the Indian subcontinent. It is expected that this knowledge will help us gain insights into how the paradigm of management of iron overload conditions in β-thalassemia is evolving in developing countries.

## 2. Physiological Iron Metabolism and Its Regulation

### 2.1. Absorption and Cellular Uptake

Dietary iron is considered to be of two types: heme iron (acquired from the hemoglobin, hemoproteins, and myoglobin of meat) or non-heme iron (acquired from iron-fortified

foods). Heme iron is reported to be more easily absorbable compared to non-heme iron [23]. During its intestinal uptake, iron is converted from its ferric state ($Fe^{3+}$) to its ferrous state ($Fe^{2+}$) by the ferric reductase duodenal cytochrome B (DYCTB) [24] at the apical side of the enterocytes facing the intestinal lumen. The proton-coupled divalent metal transporter 1 (DMT1), an iron exporter, conducts the absorption of $Fe^{2+}$ from the gut lumen into the enterocyte cytoplasm [25]. The transport of absorbed iron from the enterocytes to the systemic circulation is performed by the only identified mammalian iron exporter, Ferroportin1 (FPN1), expressed on the basolateral side of the enterocytes [26]. When FPN1 facilitates the transport of $Fe^{2+}$ to the extracellular side of the basolateral membrane, $Fe^{2+}$ is oxidized to $Fe^{3+}$ by hephaestin and ceruloplasmin (ferroxidases) for effective binding of $Fe^{3+}$ with circulatory transferrin (Tf) [27,28]. Most Tf molecules are produced by the liver [29].

The assimilation of transferrin-bound iron (TBI) into the cell is facilitated by the transferrin receptor-1 (TfR-1) present on the cell membrane of any normal cell, except on highly differentiated cells [30]. That is followed by the clathrin-mediated endocytosis of the TBI-TfR1 complex [31]. After the release of $Fe^{3+}$ in the endosome, six transmembrane epithelial antigens of prostrate 2 (STEAP2) reduces it to $Fe^{2+}$, and this $Fe^{2+}$ is transferred to the cytoplasm of the cell by Dmt1 [25]. Transferrin receptor 2 (TfR2) is the homologous protein of TfR1, and it is expressed ubiquitously on hepatocytes [32].

Species of Non-transferrin bound iron (NTBI) are also observed in the plasma, and it is considered that the major forms of NTBI include $Fe^{3+}$ bound to citrate or acetate, and its transport is facilitated by zinc transporter Zrt-Irt-like protein 14 (Zip 14), L-type and T-type calcium channels, etc. [33–35] NTBI is considered to have the most significant contribution to iron loading in the liver hepatocytes when Tf is saturated [36].

### 2.2. Storage

For the storage of iron in the cells, the major protein responsible is ferritin (Ft), which is reported to be a spherical protein nanocage of 24 subunits, consisting of heavy (Ft-H) and light (Ft-L) polypeptide chains [37]. Inside the ferritin sphere, up to 4500 atoms of iron ($Fe^{3+}$) can be stored via incorporation into a crystalline solid, called ferrihydrite $[FeO(OH)_8[FeO(H_2PO_4)]$, which restricts reactive oxygen species (ROS) formation [38,39]. Ft is contained in the cell cytosol, mitochondria, and nucleus, as well as in serum. It is observed that mitochondrial Ft (mFt) has the capability to store iron more proficiently than cytosolic Ft [40]. Ferritinophagy is the process which regulates the dissociation of iron from Ft, and it is observed that nuclear receptor co-activator 4 (NCOA4) acts as the cargo receptor by associating with the Ft-L transferring the Ft complex for degradation to the lysosome, thus making the iron stored in that Ft molecule available for biosynthetic reactions [41,42].

The hepatocytes, which comprise around 80% of the liver mass, act as the major location for the storage of absorbed iron, due to their ability to produce a large number of Ft molecules [29].

The remainder of cellular iron storage occurs within heme-containing proteins like cytochromes and Fe-S cluster-containing proteins like succinate dehydrogenase, as well as non-heme/non-Fe-S iron-containing proteins like iron- and 2-oxoglutarate-dependent dioxygenases [43–45].

Figure 1 elucidates the cellular uptake of iron by the enterocytes and the subsequent transfer of transferrin-bound iron to different cells.

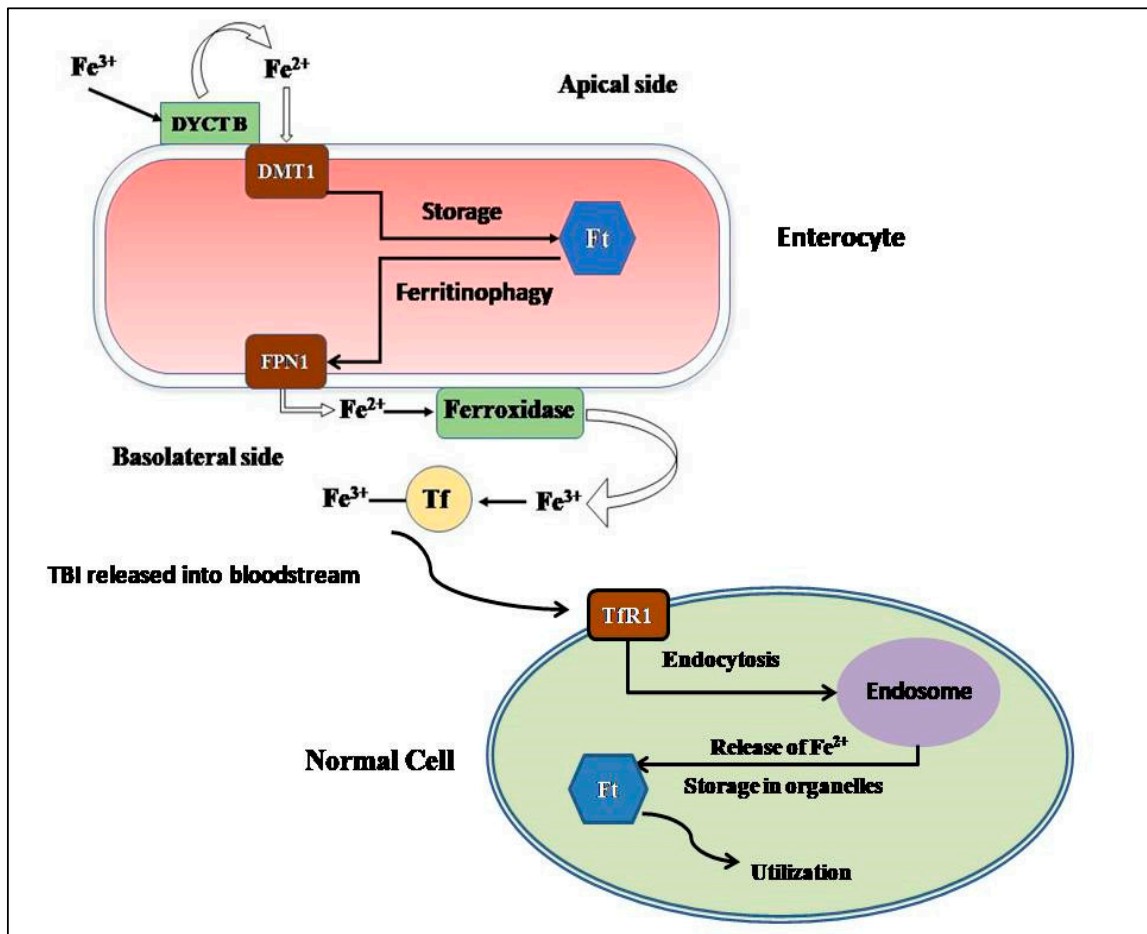

**Figure 1.** The absorption and cellular uptake of iron in enterocytes and the transport of transferrin-bound iron to other cells for utilization. (DYCTB—Duodenal Cytochrome B, DMT1—Divalent Metal Transporter 1, Ft—ferritin, FPN1—Ferroportin1, Tf—Transferrin, TfR1—Transferrin Receptor 1).

*2.3. Consumption and Recycling*

The mammalian body has a high iron requirement, with the majority of it being used for hemoglobin synthesis by the erythroblasts [46]. The mitochondrion has the most significant role in maintaining cellular iron homeostasis. Iron released from the endosomes is directed to mitochondria in one of the two following ways: (1) Iron can be transferred to mitochondria from the endosomes by a cytosolic iron chaperone protein, poly (rC) binding protein 1 (PCBP1) [47]. (2) Iron can also be delivered, without any intermediate, into the mitochondria from the endosomes via a 'kiss-and-run' mechanism, as detected in erythroid cells, since they have greater demands of iron for hemoglobin synthesis [48]. The transport of iron between the inner membranes of mitochondria is facilitated by mitoferrins 1 and 2 [49]. Inside the mitochondria, iron is utilized for the production of heme and the Fe-S clusters, which, in turn, facilitate the biosynthesis of several proteins associated with electron transfer by incorporating into them [50,51].

Senescent erythrocytes show decreased membrane flexibility, the presence of membrane phosphatidylserine, alterations on the erythrocyte solute carrier family 4 (anion exchanger) member 1 (SLC4A1), decreased sialic acid, and the cluster of differentiation 47 (CD47) antigen [52–54]. Hepatic and splenic macrophages scavenge and phagocytose these senescent erythrocytes to free iron from hemoglobin for utilization in another hemoglobin cycle [55].

*2.4. Regulation of Iron Metabolism*

The systemic regulation of the intricate metabolism of iron occurs in the mammalian body via the Hepcidin–Ferroportin axis. Hepcidin is a 25 amino-acid peptide hormone that is primarily expressed by the hepatocytes. Reports suggest that the binding of hepcidin to FPN1 on any FPN1-expressing cell types leads to rapid ubiquitination, internalization, and lysosomal degradation of FPN1, causing a disruption in the iron export from the cells and the retention of iron in them [56]. Hepcidin is expressed as a product of the *HAMP* gene, which is positioned on Chromosome 19 [57]. Erythropoiesis, anemia, hypoxia, and iron deficiency lead to decreased hepcidin production [58,59]. However, infection, inflammation, and iron overload result in increased hepcidin production [60–62]. A membrane protein, Hemojuvelin (Hjv), is reported to vitally regulate the expression of the *HAMP* gene in the liver, which acts via the Bone Morphogenetic Protein (BMP) signaling pathway [63]. A regulatory serine protease, called Matriptase-2 (encoded by the *TMPRSS6* gene), which is primarily produced by the liver, is known to cleave Hjv, and this is eventually found to impede the production and functioning of hepcidin [64].

On the cellular level, the expressions of different iron metabolism proteins, like the subunits of Ft, TfR1, and FPN1, are post-transcriptionally regulated by the association of the iron regulatory proteins (IRPs) to different highly conserved iron-responsive elements (IREs), located at the untranslated regions (UTRs) of their corresponding mRNA transcripts [65,66]. IREs can be located at either the 5′-UTR (FPN1, Ft-H and Ft-L) or the 3′-UTR (TfR1 and DMT1) [65,67–69]. IRP1 and IRP2 are RNA-binding proteins that possess the ability to sense cytosolic iron concentration and bind to their corresponding mRNA targets to modify their expression [70]. When an IRP associates with the IREs present at the 5′-UTR, the translation of the mRNA transcript is hindered and the mRNA transcript is degraded. However, when the IRP associates with the IREs present at the 3′-UTR, the mRNA transcript is stabilized and is actively translated [71].

In iron-deficient cells, IRPs bind to IREs present at the 3′-UTR of TfR1 and DMT1 mRNA transcripts, leading to the stabilization of their transcripts and subsequent translation, leading to an increase in iron import. The IRPs also bind to IREs present at the 5′-UTR of FPN1, Ft-H, and Ft-L mRNA transcripts to facilitate the degradation of their mRNA transcripts, hence facilitating the decrease in iron storage and export [72].

In iron-adequate cells, the IRPs do not bind to the IREs present at the 5′-UTRs of the FPN1, Ft-H, and Ft-L mRNA transcripts; hence, they are continuously translated. In contrast to this, the mRNA transcripts with IREs in the 3′-UTR (TfR1 and DMT1) are degraded, thus leading to decreased iron import and increased iron storage and export [71].

## 3. Iron Overload in Beta-Thalassemia

As previously mentioned, iron overload is seen as an inevitability in both TDT and NTDT β-thalassemic patients. The non-transferrin-bound plasma iron (labile iron) pool thus formed leads to the generation of ROS, which causes lipid peroxidation and leads to dysfunction in various organs, like the liver, heart, and endocrine glands [73]. Hence, β-thalassemic patients are under increased oxidative stress. The surplus iron also accumulates in the different end-organs and, in turn, leads to their subsequent dysfunction, which ultimately leads to increased morbidity [73]. However, there lies a difference in the pattern of iron loading and overloading in different organs with respect to transfusion dependency. Furthermore, studies have demonstrated that β-thalassemic individuals (carriers and patients) who have histidine to aspartic acid substitution at codon 63 (H63D) of the Hemochromatosis gene HFE have a greater propensity towards iron overloading, suggesting the modulating ramifications of H63D mutation of the HFE gene on iron metabolism [74,75].

*3.1. TDT vs. NTDT*

Around 200–250 mg of elemental iron is present in each unit of transfused packed red blood cells, while the human body is only capable of losing around 1–2 mg of iron every

day [76]. In TDT patients, around 0.3–0.6 mg/kg of transfusional iron is incorporated into the body daily, considering a monthly transfusion rate of 2 to 4 units of packed red blood cells [77].

The reticuloendothelial macrophages phagocytose the senescent transfused red blood cells; hence, iron is liberated into plasma for binding to Tf [78]. Even after the saturation threshold of the Tf molecules is reached, NTBI is transported into the cardiomyocytes, hepatocytes, and endocrine glands via the calcium channels [35]. In the cardiomyocytes, it was reported that the uptake of $Fe^{3+}$ ions was mediated via lipocalin-2 and its receptor, instead of via calcium channels [79,80]. This excess iron leads to irreversible damage in the different organs and hampers the functionality of those organs. Cardiac dysfunction, owing to iron overload in the myocardium, is one of the main comorbidities related to β-thalassemia, and it leads to nearly 71% of mortality associated with the disease [81]. In TDT patients, cardiac siderosis, which leads to arrhythmias and heart failure, along with hepatic and endocrine dysfunction, has been reported [82].

In NTDT patients, ineffective erythropoiesis triggers increased iron absorption from the intestines [83,84]. Furthermore, ineffective erythropoiesis also leads to conditions of anemia and hypoxia; thus, as previously mentioned, the hepcidin levels decline to facilitate the compensatory iron acquisition for erythropoiesis [85,86]. This results in the upregulation of ferroportin, which causes iron release from the enterocytes, as well as from the reticuloendothelial system into systemic circulation [59,87,88]. The increased iron burden leads to the deposition of iron into a variety of organs (similar to the case of TDT, but at a much slower rate), leading to oxidative damage [89]. Previously, growth differentiation factor-15 (GDF-15) and twisted gastrulation 1 (TWSG1) were implicated to have important roles in hepcidin suppression in NTDT patients. However, upregulation of these proteins was not observed in β-thalassemic mice, casting doubt over their importance [90]. It has been observed that Erythroferrone, a protein expressed by bone marrow and splenic erythroid precursors, has increased production owing to the erythropoietic stimulation for compensation for ineffective erythropoiesis. Erythroferrone is reported to contribute to iron overload, as it is seen to cause hepcidin downregulation [91].

Figure 2 shows a concise description of the pathways for iron overload in TDT and NTDT patients.

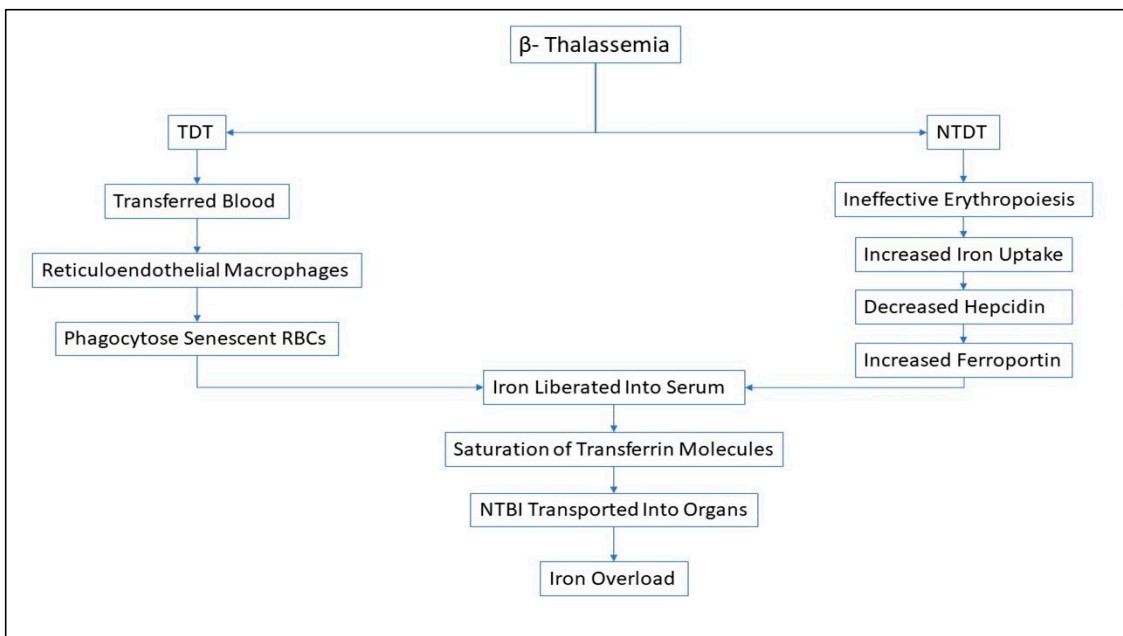

**Figure 2.** Figure showing schematic diagram illustrating the main pathways for iron overload in TDT and NTDT patients. Although ineffective erythropoiesis occurs in all β-thalassemic patients, blood transfusions lead to faster iron overload in TDT patients.

Interestingly, it has been observed in NTDT that, although the patients showed severe liver iron overload, they did not show cardiac iron overload. Hence, it has been concluded that iron overload distinctly affects the hepatocytes, instead of the cardiomyocytes, in NTDT patients [92]. TDT has been seen to be correlated with multiple complications like chronic anemia, liver fibrosis, hypothyroidism, growth retardation, diabetes mellitus, etc. It was revealed from the OPTIMAL CARE study that NTDT presents a distinct array of comorbidities that are similar to those of TDT. NTDT-related comorbidities generally include osteoporosis, hypogonadism, leg ulcers, etc., whereas TDT-associated complications like heart failure, hypothyroidism, and diabetes mellitus occurred at a lower rate in NTDT patients. Young TDT patients have been seen to develop clinical iron overload after receiving around 10–20 blood transfusions, whereas NTDT patients mostly developed iron overload slowly over the course of 10–15 years [93,94].

### 3.2. Differential Expression of Different Proteins in Iron Overload Conditions

The expression of the major apical iron transporter, DMT1, reportedly does not increase in iron overload conditions, indicating that changes in DMT1 levels are not major causes of iron overload [95]. Higher levels of the ferroxidase ceruloplasmin have also been observed in β-thalassemic patients, which possibly facilitates increased loading of iron onto transferrin and after high transferrin saturation, as well as onto albumin and citrate to enhance the formation of ROS-generating labile plasma iron [96]. As previously mentioned, there is an augmentation in FPN1 levels, due to the suppression of hepcidin, to enable increased iron transport from the cells to the plasma. Furthermore, when the iron-binding capacity of Tf is saturated and reaches its threshold due to increased iron load, NTBI increases in the plasma of β-thalassemic patients [77]. As β-thalassemia is considered to be associated with ineffective erythropoiesis, the presence of soluble Tfr1 is seen to be increased in β-thalassemic patients, probably to facilitate the transport of iron required for compensation [88]. The serum ferritin levels are greatly enhanced in β-thalassemic patients [97].

### 3.3. Detection of Iron Overload

In TDT β-thalassemic patients, iron overload can be quantified using serum ferritin, urinary iron elimination, hepatic iron content, and total iron-binding capacity of transferrin (TIBC) levels [97]. Iron toxicity is considered when the serum ferritin levels exceed 2500 ng/mL, urinary iron excretion levels exceed 20 mg/day, hepatic iron content levels exceed 440 mmol/g, and the transferrin saturation levels are greater than 75% [98]. A serum ferritin level of 1000 ng/mL indicates the threshold for starting iron chelation therapies in TDT patients [99]. In NTDT patients, a threshold value of 800 ng/mL is reported as the serum ferritin threshold level representative of iron overload in NTDT patients [100].

For evaluation of the liver iron concentrations, R2 or T2* magnetic resonance imaging (MRI) can be used. For TDT patients, if the concentration of iron in the liver surpasses 7 mg/g dry weight (dw) of liver iron concentration (LIC) values, then there is a higher propensity for iron overload, whereas LIC values greater than 15 mg/g dw increases chances of severe liver fibrosis and mortality. In NTDT patients, LIC values exceeding 5 mg/g dw is indicative of increased mortality [94,101,102].

T2* MRI is also the gold standard for detecting cardiac iron overload in milliseconds in β-thalassemic patients. The T2* is observed to become shorter when iron deposition in the myocardium increases [103]. Previous reports indicate that there is an intensifying impairment in the Left Ventricular Ejection Fraction (LVEF) when theT2* values <20 ms, and there is deterioration in the functioning of right and left ventricles when the T2* values <14 ms in β-thalassemic patients [104,105]. Severe iron overload is associated with cardiac T2* values <10 ms [106]. However, a negligible correlation has been established between cardiac T2* and serum ferritin levels, indicating that tracking ferritin levels cannot serve as a reliable indicator for assessing the cardiac condition associated with iron overload [104].

## 4. Iron Chelators

Since all β-thalassemic patients, regardless of their transfusion dependency or non-dependency, acquire iron overload, it is imperative to employ iron chelators to eliminate the excess of toxic iron in them and, thus, alleviate the symptoms of iron overload. However, when considering the utilization of iron chelators for a patient, a specific treatment regimen is chosen that benefits the individual, taking into consideration the chelating medication's long-term efficacy, safety, and cost.

The three iron chelators commonly used are Deferoxamine, Deferiprone, and Deferasirox. The chemical structures of these three iron chelators are shown in Figure 3.

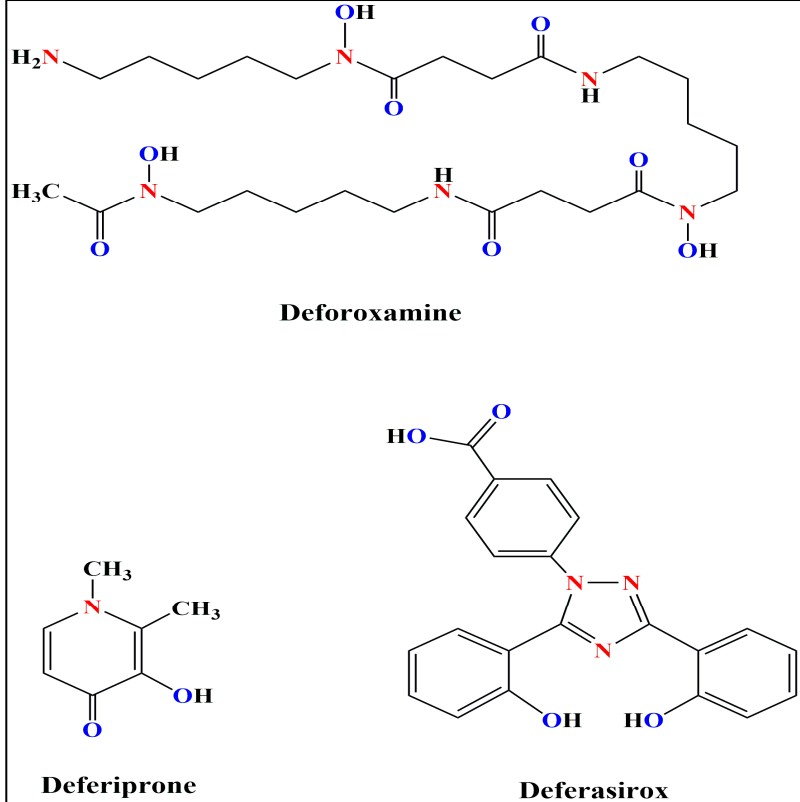

**Figure 3.** The chemical structures of the three clinically approved iron chelators used to treat iron overload.

Deferoxamine (DFO) is reported to bind to iron at a 1:1 molar ratio [107]. DFO is observed to decrease serum ferritin levels and hepatic iron overload in β-thalassemic patients [108]. However, since the plasma half-life of DFO is low, continual injections are required for iron-overloaded patients, either subcutaneously or intravenously [109].

Deferiprone (DFP) is reported to bind to iron at a 3:1 molar ratio. It is administered orally and dosages of 75–120 mg/kg/day of DFP are usually sufficient to induce a negative iron balance, inducing efficient control of cardiac as well as hepatic iron overload [110,111]. The DFP treatment regimen has a high percentage of adherence compared to that of DFO, and DFP is seen to have a better efficacy profile compared to that of DFO, as well [112].

Deferasirox (DFX) is reported to bind to iron at a 2:1 molar ratio. DFX is now used by millions of TDT patients with iron overload. DFX is administered orally, and it is seen to only increase fecal iron excretion. It has been reported that DFX lowers liver iron content and serum ferritin levels, along with increasing fecal iron excretion in iron-overloaded patients at the prescribed dosage of 10–40 mg/kg/day [113,114].

Table 1 elucidates the different characteristics of these three iron chelators.

**Table 1.** A list of different commercially available iron chelators used to treat iron overload.

| Characteristics | Deferoxamine | Deferiprone | Deferasirox |
|---|---|---|---|
| Structure | Hexadentate | Bidentate | Tridentate |
| Route of Administration | Subcutaneous or Intravenous Injections | Oral | Oral |
| Mechanism of Action | Chelates NTBI, Ft-bound iron; promotes ferritinophagy [115] | Chelates labile iron in cytosol [115] | Chelates labile iron in cytosol; increases hepcidin levels [115] |
| Route of Excretion | Biliary and Urinary [116] | Urinary [117] | Fecal |
| Adverse Effects | Hearing disorders, Growth Retardation, Lung/Renal Toxicity, Bone Abnormalities, Visual disorders, Pain at site of injection [118]. | Severe Agranulocytosis, Gastrointestinal problems, Arthiritis [118]. | Rash, Renal disorders, Gastrointestinal problems [118] |

There are currently many other iron chelators that show great promise towards mitigating iron overload, like Deferitazole and SP-420 [119,120]. Amlodipine, a calcium channel blocker, has been reported to be efficient in decreasing myocardial iron concentration and significantly elevating cardiac T2*, which indicates its role for the treatment of cardiac iron overload [121]. Furthermore, a recently discovered novel erythroid maturation agent, luspatercept, has been reported to lower the serum ferritin levels in adult β-thalassemic patients, as well as significantly reduce the number of blood transfusions (>33%) in TDT patients (BELIEVE trial) [122]. The long-term analysis of luspatercept, along with confirming the previous reports, also reported a decrease in mean daily iron chelator use in TDT patients. Furthermore, the trial has also reported a decrease in and stabilization of liver iron concentration in TDT patients after 96 weeks of luspaterecept usage [123]. All these results indicate the potency of luspatercept as an emerging efficient therapeutic agent for the treatment of iron overload in TDT patients.

## 5. Guidelines for Usage of Iron Chelators in India

Blood transfusion is reported to be one of the prevalent clinical interventions in modern medicine, alleviating the severe symptoms of patients with chronic anemias, such as thalassemia, sickle cell disease, myelodysplastic syndromes, etc., where patients require regular blood transfusions for survival or to improve their quality of life. Previous reports suggested that each milliliter of red cells is packed with approximately 0.8 mg of iron, and our physiological mechanisms limit us to effectively eliminate approximately 1–2 mg of accumulated iron per day, and only through desquamated oral and intestinal epithelia [124]. Therefore, it is evident that patients who require multiple blood transfusions are often prone to develop rapid iron overload in the body. Effective management of iron overload requires efficient monitoring of bodily iron storage. The iron status of the body is readily evaluated using different methods, as previously mentioned. Previous reports suggested that the serum ferritin level can be regarded as a reliable, cost-efficient, and readily detectable indicator of bodily iron storage. It has been extensively used to monitor the iron status of the body in recent times. With the advent of modern medicine, iron chelation has become an effective strategy to alleviate symptoms associated with iron overload in patients with transfusion-dependent chronic anemias. According to the guidelines of the Ministry of Health and Family Welfare, Government of India, once the serum ferritin levels exceed 1000 mg/L, after approximately 10–12 blood transfusions, the recommended dosages of the three iron chelators as a part of the iron overload treatment regimens are as follows:

- DFO: continuous subcutaneous injection over 8–12 h or more with the help of an infusion pump, dispersed in water; dosage, 25–50 mg/kg/day;
- DFP: orally, in 2–3 divided dosages; dosage, 50–100 mg/kg/day;
- DFX: orally, dispersed in water or juice; dosage, 20–40 mg/kg/day;
- Combination therapy: when patients no longer respond to monotherapies, it is advisable to shift to combined regimens of DFX and DFO [125].

It is absolutely essential for iron-overloaded patients to adhere to iron chelation therapies for decreased mortality, as well as decreased comorbidities. In 2017, Bhattacharyya et al. reported that DFX is found to be an effective iron chelator that can reduce serum ferritin levels efficiently in Indian HbE/β-thalassemia patients with minimal or no adverse effects [126]. However, a recent study from India has reported that non-adherence to iron chelation therapies was found in 10.7% of iron-overloaded patients. The levels of serum ferritin were reported to be exacerbated significantly in non-adherent patients in comparison to those of adherent patients, and higher rates of both cardiac and hepatic iron overload were also observed in them. It was also observed that adherence to the treatment regimen was highest with DFX, followed by DFP, and finally, DFO [127].

This is heartening to note, considering the fact DFX has been reported to have greater effectiveness in the reduction in iron overload and chelator-related side effects in TDT patients, when compared to DFP, as reported in a study conducted on children from West Bengal (Eastern India) in 2021 [128]. Furthermore, in 2021, another Indian study reported that combined oral chelation therapy conducted with DFO and DFX significantly reduces serum ferritin levels in TDT children with severe iron overload [129] These findings indicate that oral iron chelation therapy and, in particular, combined oral chelation therapy with both DFP and DFX, may yield the optimal results for the treatment of iron overload in β-thalassemic patients, although further studies in larger cohorts are warranted.

Chelator-related side effects are a common concern for patients and treating physicians. In 2021, Chandra et al. evaluated the risk of development of neutropenia between two thalassemic groups (patients on combined DFP and DFX, and patients with DFX alone). No significant correlations ($p = 0.87$) were found [130]; however, in a previous study, assessing the safety of the oral iron chelator DFP Naithani et al., in 2005, reported thrombocytopenia as a major side effect in young (<6 years) thalassemia patients in India [131]. These data indicated that, although iron chelators are essential for mitigating iron overload, the treatment regimen should be carefully optimized and monitored, especially in younger patients.

Compliance with iron chelation therapy, in spite of its absolute necessity for optimal results, is a major challenge globally. Interestingly, adherence rates to iron chelation therapies in India has been found to be significantly higher than those of the adolescents of other South Asian countries, like Malaysia (51.4%) [132]. Various reasons have been elucidated for non-adherence to iron chelation therapies, including poor family support, low family income, and side effects of the iron chelators [127].

All of these reports elucidated the present scenario of iron chelators used in treating patients with iron overload in India. Iron chelation therapy is being looked at as one of the feasible options to partially combat the complications associated with regular blood transfusions in severe chronic cases of anemias in India. However, further studies are required to determine the most effective and safe usage of these drugs in patients with heterogeneous clinical symptoms associated with iron overload. Until such time, strict monitoring is required to administer the optimal dosages of these iron chelators in different clinical settings in India.

## 6. Conclusions

Iron overload is an indispensable comorbidity associated with β-thalassemia. Thus, iron chelation therapy is an absolute necessity when it comes to the holistic treatment of β-thalassemic patients. Despite the high adherence rate mentioned in the aforementioned study, it is difficult to quantify the adherence to iron chelation therapies all over India. More research is crucial to understand the entire picture of iron chelation regimens and

the adherence to them across the whole demographic of India. The β-thalassemia trait is unequally distributed across the Indian subcontinent, so it is necessary to provide adequate treatment to all iron-overloaded patients or to those who have a higher propensity to develop iron overload. Hence, affordable healthcare options are very necessary for the effective clinical management of iron overload, as the medications are currently quite costly, and might benefit a higher number of people affected with β-thalassemia all over India.

**Author Contributions:** S.B., M.R. and N.C. were involved in original draft preparation; P.C.S., T.K.D. and N.C. have reviewed and edited the manuscript. All authors have read and agreed to the published version of the manuscript.

**Funding:** This study was supported by the Department of Biotechnology (DBT), Ministry of Science and Technology, Government of India (Project Title: "Micro-RNA based reprogramming of fetal hemoglobin in beta-thalassemia", Sanction no: BT/PR32054/MED/97/454/2019).

**Institutional Review Board Statement:** Not applicable.

**Informed Consent Statement:** Not applicable.

**Data Availability Statement:** Not applicable.

**Acknowledgments:** Subhangi Basu acknowledges the University Grants Commission (UGC), the Government of India, and New Delhi for providing a research fellowship. The authors thank the Indian Institute of Technology Kharagpur for providing infrastructural support. The authors acknowledge the Department of Biotechnology (DBT), Ministry of Science and Technology, Government of India for providing funding.

**Conflicts of Interest:** The authors declare no conflict of interest.

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
