# Peer review of "Understanding the Intricacies of Iron Overload Associated with β-Thalassemia: A Comprehensive Review"

_thalassrep, doi:10.3390/thalassrep13030017_

Round 1
Reviewer 1 Report
I think it is useful to make a report to new treatment for thelassemia like luspatercept and the effect of it on iron overload and if it is applicable in India
Author Response
Thank you for your review and suggestion.
As per your recommendation, we have added the following part to Section 4. Iron Chelators-
Further, a recently discovered novel erythorid maturation agent, luspatercept, has been reported to lower the serum ferritin levels in adult β-thalassemic patients as well as significantly reduce the number of blood transfusions (>33%) in TDT patients (BELIEVE trial) [122]. The long-term analysis of luspatercept, along with confirming the previous reports, also reported a decrease in mean daily iron chelator use in TDT patients. Furthermore, the trial has also reported a decrease anf stabilization of liver iron concentration in TDT patients from 96 weeks of luspaterecept usage [123]. All these results indicate the potency of luspatercept as an emerging efficient therapeutic for the treatment of iron overload in TDT patients.
Reviewer 2 Report
This well written review described the physiologic iron metabolism and its regulation at cellular level and also explored the mechanisms of iron overloading in TDT and NTDT b-thalassemic patients. It also gave information about the different methods for the quantification of iron overload and about the use of different iron chelators in India. The review is well structured and concepts are generally well explained.
I strongly suggest to revise the manuscript, addressing these minor issues:
Page 2 lines 54-57: The sentence is too long and needs an English revision. Please, rescale it.
Page 2 line 59: The procedure of hematopoietic stem cell transplantation is not performed to alleviate the disease symptoms but to resolve the pathology. Please, correct the sentence.
Page 6 line 212: Please, give a more recent reference.
Page 7 lines 233-235: It was demonstrated that in thalassaemic patients (not only NTDT but also TDT) cardiac T2* values do not correlate with serum ferritin concentration and liver iron concentration. The relationship between cardiac T2* values and iron balance is quite complicated because the mechanisms and kinetics of cardiac iron uptake and clearance differ from the liver. Please, rescale the sentence considering this comment.
Page 8 lines 258-261: this small paragraph is not about the “detection of iron overload”, I think that it could be moved to another part of the manuscript.
Page 8 line 262: Hepatic iron content is not a blood marker, please correct the sentence.
Page 8 lines 279-280: I strongly suggest to expand the sentence, explaining why cardiac iron is usually not related to serum ferritin levels, considering that ferritin is also a well-known inflammatory marker.
Page 9 lines 298-301: I suggest to expand the paragraph about deferiprone because its efficacy is not described. The reference n.110 was published in 2002, about 20 years ago. Please, consider a more recent publication.
I suggest the revision of the manuscript by an English native speaker.
Author Response
Thank you for your review and suggestions.
As per your recommendation, we have made the following changes-
Comment: Page 2 lines 54-57: The sentence is too long and needs an English revision. Please, rescale it
Response: The sentences have been modified as follows: Due to the quantitative reduction in β-globin, particularly in individuals with TM and TI, excess accumulation of α-globin chains in erythroid precursors has been reported. This causes globin chain imbalance, resulting in a state called ‘ineffective erythropoiesis’. Under such circumstances, in an effort to enhance erythrocyte production (red blood cells), the developing nucleated erythroid cells undergo premature apoptosis as a means to restore equilibrium.
Comment: Page 2 line 59: The procedure of hematopoietic stem cell transplantation is not performed to alleviate the disease symptoms but to resolve the pathology. Please, correct the sentence
Response: The sentence has been corrected as follows: This ultimately leads to chronic hemolytic anemia, requiring regular blood transfusions and other therapeutic approaches like iron chelation therapies, fetal hemoglobin upregulation, etc. to alleviate their disease symptoms as well as other clinical approaches like hematopoietic stem cell transplantation, etc. to resolve the disease pathology.
Comment: Page 6 line 212: Please, give a more recent reference
Response: The following reference has been added: V. Russo, A. Rago, A. A. Papa, and G. Nigro, “Electrocardiographic Presentation, Cardiac Arrhythmias, and Their Management in β-Thalassemia Major Patients.,” Ann Noninvasive Electrocardiol, vol. 21, no. 4, pp. 335–42, Jul. 2016, doi: 10.1111/anec.12389.
Comment: Page 7 lines 233-235: It was demonstrated that in thalassaemic patients (not only NTDT but also TDT) cardiac T2* values do not correlate with serum ferritin concentration and liver iron concentration. The relationship between cardiac T2* values and iron balance is quite complicated because the mechanisms and kinetics of cardiac iron uptake and clearance differ from the liver. Please, rescale the sentence considering this comment
Response: This part has been explained in the first part of Section 3.1. TDT vs NTDT and Section 3.3. Detection of Iron Overload.
Comment: Page 8 lines 258-261: this small paragraph is not about the “detection of iron overload”, I think that it could be moved to another part of the manuscript
Response: This part has been moved to Section 3. Iron Overload in Beta-Thalassemia.
Comment: Page 8 line 262: Hepatic iron content is not a blood marker, please correct the sentence
Response: The sentence has been corrected as follows: In the TDT β-thalassemic patients, the iron overload can be quantified using serum ferritin, urinary iron elimination, hepatic iron content, and total iron binding capacity of transferrin (TIBC) levels.
Comment: Page 8 lines 279-280: I strongly suggest to expand the sentence, explaining why cardiac iron is usually not related to serum ferritin levels, considering that ferritin is also a well-known inflammatory marker
Response: The sentence has been modified as follows: However, a negligible correlation has been established between cardiac T2* and serum ferritin levels, indicating that tracking ferritin levels cannot serve as a reliable indicator for assessing the cardiac condition associated with iron overload
Comment: Page 9 lines 298-301: I suggest to expand the paragraph about deferiprone because its efficacy is not described. The reference n.110 was published in 2002, about 20 years ago. Please, consider a more recent publication
Response: The paragraph has been expanded as follows: Deferiprone (DFP) is reported to bind to iron at a 3:1 molar ratio. It is administered orally and dosages of 75-120 mg/kg/day of DFP are seen to be usually sufficient to induce a negative iron balance, inducing efficient control of cardiac as well as hepatic iron overload[110][111]. DFP treatment regime has a high percentage of adherence rate compared to that of DFO and it is seen to have an efficacy profile compared to that of DFO as well [112].
(A recent reference has also been added.)
Reviewer 3 Report
This is an extensive and detailed review of iron overload and chelation therapy in thalassemia. As a literature review, it is comprehensive and detailed in explaining the pathophysiological mechanisms.
However, the abstract promises usage data in India that is not reported into the text. I would be very interested in seeing some data on the Indian reality rather than just a nice overview as many are available in the literature
Abstract must be modified.
Overall the paper is too long and can be significantly reduced without losing any important aspect.
English language is fine
Author Response
Thank you for your review and suggestions.
As per your recommendation, we have modified the Abstract. Furthermore, we have added more data in the Indian context in Section 5. Guidelines for Usage of Iron Chelators in India, as follows-
This is heartening to note, considering the fact DFX has been reported to have greater effectiveness in reduction of iron overload and chelator-related side effects in TDT patients, when compared to DFP, as reported in a study conducted on children from West Bengal (Eastern India) in 2021[128]. Furthermore, in 2021, another Indian study reported that combined oral chelation therapy done with DFO and DFX significantly reduces serum ferritin levels in TDT children with severe iron overload[129] These findings indicate that oral iron chelation therapy and in particular combined oral chelation therapy with both DFP and DFX, may yield the optimal results for the treatment of iron overload in β-thalassemic patients, although further studies in larger cohorts are warranted.
Chelator-related side effects are a common concern for patients and treating physicians. In 2021, Chandra et al. evaluated the risk of development of neutropenia between two thalassemic groups (patients on combined DFP and DFX and patients with DFX alone) were observed. However, no significant correlations (p=0.87) were found[130]. Although, a previous study, assessing the safety of oral iron chelator DFP, Naithani et al. in 2005 had reported thrombocytopenia as the major side-effect in young (<6 years) thalassemia patients in India. [131]. These data indicated that although iron chelators are essential for mitigating iron overload, the treatment regime should be carefully optimized and monitored, especially in younger patients.
Compliance to iron chelation therapy, inspite of its absolute necessity for optimal results, is a major challenge globally. Interestingly, adherence to iron chelation therapies in India has been found to be significantly higher than that of the adolescents of other South Asian countries like Malaysia (51.4%)[132]. Various reasons have been elucidated for the non-adherence to the iron chelation therapies including poor family support, low family income as well as side effects of the iron chelators[127].
Round 2
Reviewer 1 Report
it is a well written manuscript
Reviewer 2 Report
The authors addressed all the issues.
Reviewer 3 Report
no further questions